# Dual Vocational Education and Training Policy in Andalusia: The Nexus between the Education System and the Business Sector in the Higher-Level Training Cycle of Early Childhood Education

**Magdalena Jiménez Ramírez** * , **Rocío Lorente García and Juan García Fuentes**

Department of Pedagogy, Faculty of Education, University of Granada, 18071 Granada, Spain; rociolg@ugr.es (R.L.G.); garciafuentesj@ugr.es (J.G.F.)
* Correspondence: madji@ugr.es

**Abstract:** Dual vocational education and training (dual VET) is a recent policy aimed at establishing strong connections between the education system and the business sector. It serves to ensure the continuity of training, reduce early school leaving, and actively involve the business sector in the training and qualification of students, known as apprentices. Consequently, this approach has a favourable impact on their successful integration into the labour market upon completion of the dual training program. In this study, we investigate the implementation of dual VET in the autonomous community of Andalusia, Spain, focusing on the higher-level training cycle of early childhood education, which falls under the professional family of sociocultural and community services. Through a thorough analysis of interview accounts involving various stakeholders, we shed light on the outcomes of this policy's implementation. The findings suggest that this emerging policy may have a positive impact on the employability of young individuals by enabling in-company training, which provides them an opportunity to showcase their vocational skills and to combine practical experience with theoretical knowledge. Work mentors are identified as essential contributors to the success of in-company training, as they help foster the necessary capabilities to ensure that dual VET is perceived as a comprehensive training experience rather than as just work experience.

**Keywords:** dual vocational education and training; early school leaving; educational policies; early childhood education; labour market insertion



## 1. Introduction

Vocational education and training (VET) has been the area of the Spanish educational system that has shown the slowest progress towards modernization in the country's contemporary history, but, in recent years, it has garnered the highest levels of media and political attention. This heightened focus can be attributed, in part, to the influence of various international organizations, such as the European Union (EU) (Martínez-Izquierdo and Torres Sánchez 2022b) and the Organisation for Economic Co-operation and Development (OECD) (Valiente and Scandurra 2017). However, the Spanish situation does not project an optimistic outlook, especially in a country with fewer employment opportunities, due to the fact that it continues to be characterized by less dynamic labour markets, where youth unemployment rates are among the highest among developed countries as a whole.

The modernization discourse surrounding Spanish VET has been centred on promoting the concept of "dual training" and advocating for its urgent implementation across Spain and its regions. Spain has become an importer of dual VET as a "successful practice" and a transfer policy aimed at transforming and demanding an efficient VET system. Furthermore, the current post-COVID-19 landscape emphasizes the need for "resilience", suggesting a market-oriented dualization of VET, which may inadvertently lead to a decline

in the quality of lower-ranked courses (Esmond and Atkins 2022). For Spain, this presents additional challenges and possibilities in pursuing educational reforms, given the complex and novel nature of dual VET within the country's educational policy landscape.

The influence of dual VET is enhanced by its role as a key policy for addressing the challenges posed by early school leaving (Carrasco et al. 2021). It provides young individuals with a continuing education that links school with training in a workplace setting. This enables them to obtain higher qualifications and specialization, potentially leading to improved employment opportunities and enhanced employability (Rodríguez Crespo and Ramos Díaz 2016; Consejo Económico y Social de España 2023).

As a policy with a strong "Europeanization influence in the discourse of social actors" (Martín Artiles et al. 2019, p. 146), dual VET aims to bridge the gap between the education system and the labour market, thus preventing early school leaving (Cedefop 2016a, 2016b) and facilitating the better integration of young people into the workforce, thereby increasing their professional qualifications and a producing a more skilled workforce through work-based learning (Rego Agraso et al. 2015; Psifidou and Ranieri 2020).

This article examines specific features of the implementation of dual VET in Spain, with a particular focus on its development in Andalusia within the context of the Higher Technician in Early Childhood Education Program. This study contributes to the ongoing discourse surrounding the reduction in early school leaving rates through the adoption of this vocational training approach. It delves into various aspects, including the different pathways chosen by young students entering a dual VET, the influential factors affecting their access to such training, and the experiences of the stakeholders involved in the implementation of dual VET. Notably, it highlights the significant advantages of acquiring practical training in a company setting, as expressed by those engaged in the process.

These perspectives offer valuable insights for understanding and assessing the potential impact of dual VET as an effective policy to combat early school leaving while simultaneously increasing the chances of securing employment in the labour market, thereby addressing the issue of youth unemployment rates. All of this is derived from the influence that international organisations (OECD and EU) have been exerting on the vocational training systems of their member countries to understand, from a transnational perspective, the role of institutions and systems, and the relationship of VET with the labour market (Cedefop 2022).

## 2. Theoretical Framework

### 2.1. The Transfer of Dual VET Policy: Commitment to the Continuity of Youth Training, Apprenticeships, and Employability

Dual VET in Spain emerged as a recently developed policy under Royal Decree 1529/2012, issued on 8 November, which aimed at facilitating the gradual implementation of dual vocational training across the country. Interestingly, this policy was introduced by royal decree, without specific demands from employers, trade unions, or the education system (Marhuenda-Fluixá et al. 2017). This regulation decreed an unconventional training system, transferring models from other education systems, but without a study of the Germanic model of dual vocational training (Echeverría Samanes 2016).

This dual training policy has experienced a significant boom in order to face the challenges arising from the economic crisis, and is focused on enabling vocational training for young people, thus avoiding early school leaving (Cedefop 2016a, 2016b; Consejo Económico y Social de España 2023) and allowing training programs to connect with the demands of the labour market to provide professional qualifications for young people to increase their employability (Rego Agraso et al. 2015; Šćepanović and Martín Artiles 2020; Valiente and Scandurra 2017). Moreover, the need to implement this policy has been encouraged by the EU and the OECD, institutions that have actively promoted the benefits of dual vocational training and the transfer of policies from one country to another (Li and Pilz 2021), which recognise the importance of work-based learning (Psifidou and Ranieri 2020).

The EU has made it possible for dual vocational training policies to be implemented in all its member countries by providing financial and organisational support for their development, although with an uneven application depending on the organisational structures of each country, which do not always provide incentives to companies for training (Šćepanović and Martín Artiles 2020). The research by Martínez-Izquierdo and Torres Sánchez (2022a) corroborates the European Union's commitment to the transfer of this policy of transnational attraction based on work-based learning as a pillar that articulates apprenticeship, evidencing the "agency" role developed by the EU to disseminate dual vocational training as a "model of good practice" (p. 15).

A literature review study by Valiente and Scandurra (2017) on the implementation of dual apprenticeships in OECD countries argues that states are advocating for dual apprenticeships as a model that can be transferred to different national and local contexts, especially given the low levels of youth unemployment that dual apprenticeships bring. However, the research highlights the problematic nature of this transfer, as different actors are involved in the governance of the policy, coupled with the challenge of attracting both employers and young learners to dual apprenticeships.

In this sense, it is the German model of dual vocational training that enjoys the greatest international recognition. It is characterised by combining the training of apprentices between part-time vocational schools (Berufsschule) and companies (Alemán Falcón 2015), alternating between the two from a coordinated perspective. This organisational and training structure offers social and economic advantages, by providing skilled labour to the business fabric (Alemán-Falcón and Calcines-Piñero 2022) and facilitating the transition from school to the world of work (Cedefop 2020), although this training takes place within a system that involves a "vertical stratification" of dual vocational training, which generates differences in job opportunities linked to one's level of study and social origin (Protsch and Solga 2015).

However, although the German model of dual vocational training has been presented as a referent model for implementation in other countries, the literature shows the existence of different models of vocational training (Cedefop 2021) and dual training in the European framework, which are not limited to the Germanic model (Homs 2016). It can be said that all EU countries have dual vocational training systems, albeit with heterogeneity among them in terms of their approach, conception, and implementation (Cedefop 2018).

A distinction is made, with many nuances, between those that base vocational training in the company (*Firm-based*), such as in Germany or Austria, and those in which training takes place in vocational training centres, although with periods of work experience in companies, such as in Spain, and are the most widespread in Europe (*School-based*). Less widespread intermediate systems (*Hybrid*) have also been described, which carry out training programmes in companies and are usually associated with social inclusion policies (Cedefop 2018; Consejo Económico y Social de España 2023). (See Table 1).

**Table 1.** Apprenticeship training systems in Europe.

| Firm-Based Scheme | Hybrid Scheme | School-Based Scheme |
|---|---|---|
| • Learning aimed at providing competencies and skills for specific occupations recognized by public administrations.<br>• The learning system is independent of the educational system.<br>• The learning system is structured into programs that are uniformly applicable in all companies.<br>• Qualification: qualified worker.<br>• Countries: Germany, Austria, Denmark, or Norway. | • The learning is aimed at enabling the young population to attain qualifications that facilitate access to the job market.<br>• Associated with social inclusion policies. System with its own learning programs, independent of the education system, although they are somewhat unstructured.<br>• The system is regulated to ensure consistent delivery within companies.<br>• Qualification: a worker with professional competencies.<br>• Countries: Belgium (French-speaking) and Cyprus. | • The learning system is part of the vocational education system.<br>• Provides qualifications equivalent to in-person vocational training.<br>• Training within the company is not always regulated and can vary between them. Qualification: an individual with professional competencies. Countries: Netherlands, France, Belgium (Flemish-speaking), Spain, Italy, Portugal, or Greece. |

Source: Taken, adapted, and translated by the authors (Consejo Económico y Social de España 2023, p. 25).

Specifically, in the case of Spain, the German dual model has been the training and learning structure chosen to implement and transform traditional vocational training which, in addition to the commitment of the EU, the OECD, and the Ministry of Education and Vocational Training itself, also includes the Bertelsmann Foundation and the Alliance for Dual Vocational Training as influential actors in this transformation. The first regulatory document approved was Royal Decree 1529/2012, which regulated the development of the dual system in terms of training actions. The objective of this was the professional qualification of workers, who would alternate between working at a company and completing training activities within the framework of the vocational training system, establishing a structure with two subsystems (Gamboa Navarro and Moso Díez 2020). It configured a dual VET option; one offered by the education system, with its intermediate and/or higher-level training cycles, and another option of vocational training for employment, through an accreditation with certificates of professionalism.

Therefore, it opted for a model based on the development of dual training projects, always authorized by the educational administration, with the co-participation of educational centres and labour entities (Confederación de Empresarios de Andalucía 2017). Similarly, one of its aims was to motivate students towards vocational training in order to reduce early school leaving (Royal Decree 1529/2012 2012, article 28, 2 b.).

One year later, in 2013, the dual mode of vocational training was regulated in a central Spanish law, the Organic Law 8/2013 for the Improvement of Educational Quality (known in Spain as LOMCE). This law incorporated basic vocational training into compulsory education, which has also provided places for students who opt for the dual modality. One of the aims of this law was to boost the number of students who complete vocational training in order to improve employability, revitalise vocational apprenticeship, and lead to a reduction in early school leaving. This system aimed to mirror and transfer the German training models to Spain, given the success it has had in lowering youth unemployment rates (Marhuenda et al. 2016). Dual VET is defined in LOMCE as a set of training actions and initiatives that, with the co-responsibility of companies, aim at the professional qualification of people, harmonising teaching and learning processes between educational centres and workplaces (Organic Law 8/2013 2013, article 42, p. 34).

During the course of this decade, the uptake of dual vocational training has been increasing, although statistical data indicate that the percentage of students enrolled is not significant. For the 2019/2020 academic year, out of the entire number of vocational students, dual VET students only represented 5.4% in the higher level, 3.6% in the intermediate level, and 1.1% in basic vocational training in the on-site mode. Without differentiating by stage, dual VET students represented only 4.2% of the total (Ministry of Education and Vocational Training 2021). Even so, VET courses in general have increased the most, particularly the higher-level courses, which are expected to increase by 6.9% for the 2020/2021 school year (ibid.). According to the Consejo Económico y Social de España (2023), there has been robust growth in the enrolment of students in the dual mode of education. However, it is crucial to note that the proportion of students opting for this mode remains relatively low, despite experiencing a particularly significant increase in Andalusia.

Consequently, the challenges in bridging the gap between training and work pose a formidable task for the Spanish education system. This causes disadvantages, in terms of the adaption to changes and labour demands, mismatches between supply and demand in the labour market environment, and limitations to training qualified people from professional practices to respond to the changing needs of the labour environment (Confederación de Empresarios de Andalucía 2017; Martínez García 2015). All these challenges have arisen in a context that is characterised by high youth unemployment for those under 25 (30.03% (first quarter 2023) in Spain vs. 37.97% in Andalusia) (Instituto Nacional de Estadística 2023).

To address these challenges and accommodate more flexible training in line with current labour demands, vocational training stands as a strategic pillar for converging the objectives of training, economic needs, and business requirements. It contributes to

enhancing employability by offering an alternative to university studies, and this vision is reinforced by the Next Generation EU Funds and active employment policies in Spanish regions. These initiatives have introduced components aimed at labour reform to boost employability, enhance professional performance, and enable retraining tailored to new labour realities (Molina Belmonte et al. 2022).

Likewise, national plans have also aligned with this direction. The First Strategic Plan for Vocational Training in the Education System 2019–2022 (Ministry of Education and Vocational Training 2019) and the Plan for the Modernisation of Vocational Training (Ministry of Education and Vocational Training 2020) were committed to improving professional skills and competences to strengthen human capital and foster employability through training that meets the needs of the productive sector. To ensure the implementation of quality vocational training, a medium- and long-term approach involving companies and a study of the most demanded profiles with higher participation is crucial to avoid counterproductive effects and mismatches between supply and demand. The economic reality is also considered a determining factor for success, requiring a context of sustained and robust growth (Zúñiga Guevara and Soriano Ayala 2019).

The recent approval of Organic Law 3/2022, of 31 March, On the Organization and Integration of Vocational Training, further advances the goal of transforming vocational training into a means of social and economic transformation. The law consolidates the dual training modality to raise the level of professional qualification and requalification throughout individuals' lives. It unifies educational vocational training and training for employment into a single system, ensuring that all vocational training will be dual. This shift fosters stronger connections with the work environment, harmonizing teaching and learning processes between training centres and companies, and ultimately enhancing employability. It addresses issues such as reducing early school leaving and discrepancies in the labour market, particularly relevant after the economic crisis and the COVID-19 pandemic.

This training approach also seeks to ease the transition from other stage of the education system to vocational studies, expanding the number of young people with post-compulsory qualifications and sustainable employability prospects. Furthermore, vocational training has proven to be a valuable option in curbing early school leaving (Organic Law 3/2022 2022). European policies in recent years highlight the role of vocational training in preventing and countering early school leaving, and its positive impact on attracting, retaining, and integrating young people with diverse skills and learning experiences (European Commission/EACEA/Eurydice/Cedefop 2014; Cedefop 2016a, 2016b).

Supporting this perspective, studies have demonstrated that vocational training graduates experience lower unemployment rates, with dual training notably enhancing job placement opportunities through the strengthening of links between educational centres and the productive sector (Alemán Falcón 2015; Tolino Fernández-Henarejos 2015; Moldes Farelo and Molina García 2020; Gamboa Navarro and Moso Díez 2021).

This approach also establishes a model of vocational training, with the recognition and accreditation of competences and vocational guidance based on training pathways. These pathways are structured into five ascending grades that establish an innovative system of vocational training grades (A, B, C, D, E), according to their duration, moving from micro-training (grade A) to degrees and specialisation courses (grades D and E). The training progression to obtain accreditation, certification, and qualification includes: (A) partial accreditation of competence, (B) certificate of competence, (C) professional certificate, (D) training cycle, and (E) specialisation course-master.

Dual VET will be developed within grades C and D and, where appropriate, E, and will include a period of compulsory training in companies. This training will be divided into two types: general vocational training and intensive vocational training. In the general option, the time spent at the company is between 25% and 35% of the total duration of the training, and the company makes a commitment to participate in up to 20% of the content and learning outcomes of the curriculum. In the intensive option, in-company training makes up 35% of the total duration of the course, and the company collaborates with the

training centre in the development of more than 30% of the curriculum. In addition, these training placements include a training contract between the student and the company under the terms determined by labour legislation, and must be paid within the framework of a training contract. All persons obtaining a vocational training qualification, then, will have spent a training period at a company. Likewise, in a society that demands professionals with increasingly higher levels of qualification, it is imperative to address the limited opportunities for young individuals who leave school early, i.e., the labour market will not be able to take on workers with few or scarce qualifications (Martín Rivera 2016).

### 2.2. Dual VET in Andalusia: Configuration, Implementation, and Concretion of the Higher Technician in Early Childhood Education

The decentralisation of education in Spain means that each autonomous community (in fact a quasi-federal region) regulates the conditions for the development of dual training, which allows for particularities and differences in the interpretation, development, and implementation of the approved regulation. In this sense, the research considers "situated learning" (Hoffmann and Evans 2004) as a pedagogical approach to study how the implementation of dual vocational training develops depending on based on the autonomous community. The study of this policy involves situating the learning that young people acquire at school and at work in order to find out whether it contributes to improving their training and, therefore, to reducing early school leaving, as well as to their prospects for obtaining a job that will lead to the socio-occupational integration of young people.

The experimental development of this training model began in Andalusia (a region of southern Spain) in 2013 and involved a total number of 12 training projects, involving 11 public schools, 207 participating students, and 87 collaborating companies. After the first experimental phase and in the following years, dual vocational training has grown and seems to be consolidating as the training modality for the future (Regional Department of Education and Sport 2021). In the case of the autonomous community of Andalusia, dual training places are offered in basic, intermediate, and higher vocational training cycles, which have a duration of at least 2000 h over an average of two academic years (Figure 1).

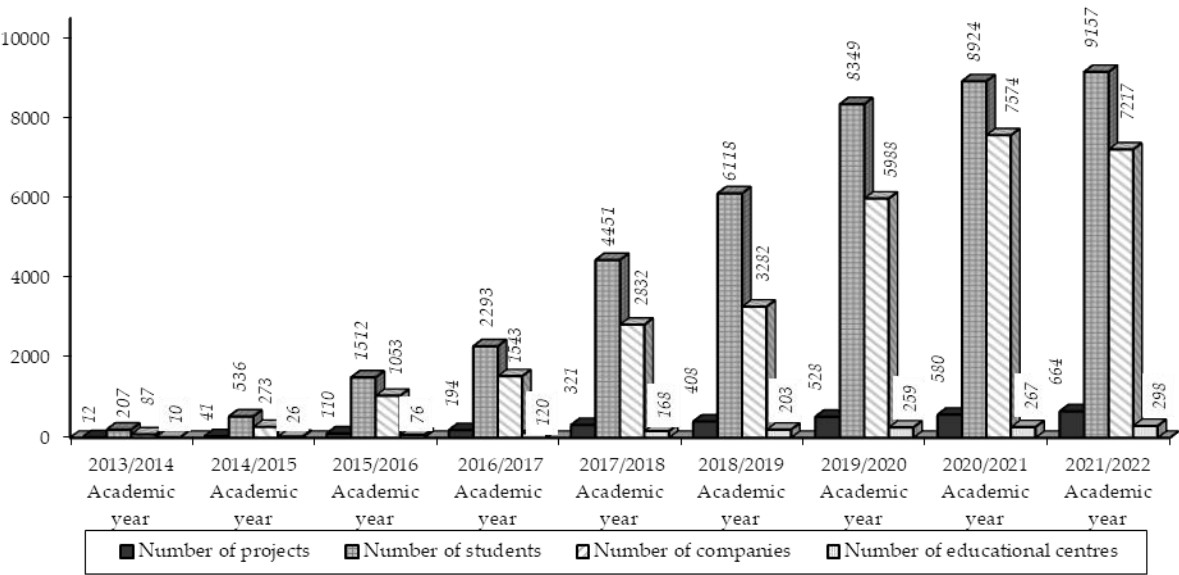

**Figure 1.** Evolution of dual VET in Andalusia. Source: Own elaboration based on data from the Regional Department of Education and Sport.

In order to offer dual training, there is an annual call announced by the regional government, when the management of the schools, their teaching staffs, and the companies that decide to do so, draw up and present a dual training project for assessment. These dual training projects are characterised by their flexibility, taking into account the idiosyncrasies

of the Andalusian business environment, which has a majority presence of small and micro companies, and a smaller presence of large companies. Another feature of the projects are their adaptability to a dual theoretical–practical vocational training model between the schools and companies, considering the socio-productive and business fabric where the educational centre is located, as well as the key role of market prospecting (Confederación de Empresarios de Andalucía 2017). Therefore, each training project develops a dual offer for a specific school, with a programme of theoretical–practical activities that is different from other dual projects in the same professional family.

These dual projects in Andalusia combine the training, apprenticeship, and certification of students through coordinated processes between the schools and work centres. Students receive theoretical and practical training in a concrete work environment, which allows them to obtain practical experience in the profession, and maximises their chances of accessing the labour market by acquiring professional skills in the workplace. Thus, in this research we consider the connection between school and work by understanding the learning process as a combination of theory and practice in training for the development of general and professional competences. Hence, the focus is on connective learning situations (Stenström and Tynjälä 2009) involving individuals, working communities, and organisations and institutions, which constitute learning spaces in the workplace from this dual vocational-training-policy-based approach (Evans and Rainbird 2002).

The key to managing this training process are the figures of the dual tutor at the school and the company tutor, who agree on a dual training plan for the apprentices to develop at the workplace, in coordination with the dual coordinators and the teaching staff at the educational centre. In this dual mode, practical training at the company is carried out in two different ways: either young students can combine their theoretical training at the school with practical training at the workplace every week, or they can take an initial period of training at the school and then carry out the dual training at the work organisation.

In order to find out about the progress of this type of vocational training in Andalusia, we focus on the study of the implementation of dual training for the grade of Higher Technician in Early Childhood Education, which is a two-year course that belongs to the professional family of sociocultural and community services, the professional family with the lowest rate of early dropout in the academic year 2019/2020 (Consejo Económico y Social de España 2023, p. 64). We are interested in knowing the role attributed to dual vocational training in our educational, social, and economic contexts; the aspects related to management and the dual training processes; and the coordination with the work centres. We also wish to assess the challenges that have derived from this training policy, particularly in relation to the acquisition of the competences that qualify trainees for labour market insertion, thus avoiding early school leaving among young people.

## 3. Methods

This research aims to explore the transitions of young people to dual vocational training programs and assess the extent to which this policy meets the expectations for the continuous training and enhanced socio-labour inclusion of the participating youth. The primary focus is on the practical training provided in the workplaces, which advantageously positions them for professional qualification and practice.

The development of this objective was contemplated in a research project funded by the Ministry of Science, Innovation and Universities, "Challenges of the implementation of dual training in Spanish vocational training" (RTI 2018-101660-B-100), in which a qualitative study approach was proposed (Ispizua and Lavía 2016). This research work has been elaborated upon to consider the Code of Good Research Practices of the University of Granada (Universidad de Granada 2019), respecting the ethical criteria that should be applied to all scientific practice.

As this was a qualitative type of research study, it was assumed that the "foundations of truth and knowledge can be defined by an infinite variety of paths—as many as the rational process in which the researcher is involved in, or the results of the empirical

observation undertaken" (Moral Santaella 2006, p. 156). In her review of the international literature on validity criteria in present-day qualitative research (Moral Santaella 2006, p. 56), and basing her views on those of Lincoln and Guba (1999) and Smith and Deemer (2000), this author describes how "truth and valid knowledge are built upon consensus among the members of the community (...), [where] true knowledge is accessed through dialogue, and therefore we move from an objective validity to a communal validity through the reasoning of the participants in the discourse". Moral Santaella (2006) establishes that "if the ultimate goal of social enquiry is directed toward the transformation of society itself, then the only valid criteria are those that lead us to that transformation" (p. 157), specifying that "the criteria for transformation lie within a process of dialogue" (ibid.).

The participating sample consisted of seven educational centres—public, private subsidised, and private—in the autonomous community of Andalusia. The number of participating centres is justified by the limited number of places and educational centres that have developed dual vocational training for the higher vocational training cycle in early childhood education in Andalusia, given that its implementation is recent and has been carried out unevenly, depending on the professional families and the needs and concerns of the centres that provide vocational training.

Our research sampling followed a non-probabilistic method. Specifically, we selected cases that were typical and considered the most significant for the aim of the study (Rubio and Varas 2010); significancy (ibid., p. 429) was one of the keys for the selection of the most representative persons for the aims of this research and considering the study population. As our qualitative sampling had no pretension to be statistical, but instead typological and having socio-structural representativity (Valles 2009, p. 68), we selected key informants (Rubio and Varas 2010, p. 430) of interest for the research.

Specifically, twenty-nine informants participated in the study: headteachers of educational centres, directors of dual vocational training partner companies, coordinators of dual vocational training at the schools, tutors at the schools, work tutors, teachers who provide dual vocational training, and ten young students in dual vocational training (Appendix A). Knowing the accounts of the different experts involved in the governance and implementation of dual vocational training (Sanz de Miguel 2017) is essential because, although it is a policy designed for young people as the beneficiaries, they are not the only actors involved in the coordination, management, development, and possible success of the policy.

To advance the research, we used a semi-structured interview, with an interview script (Valles 2009) that guided the in-depth interviews with the actors involved in the development of the policy. Following the process suggested by Rubio and Varas (2010), a semi-structured interview was designed and applied, organised around several thematic blocks related to the implementation of dual vocational training in Andalusia: a. the integration of dual vocational training in schools; b. the involvement of the business sector in dual training and access to the labour market; c. the governance of dual vocational training; d. the assessment of the development of the dual training subsystem; e. international models of dual vocational training; and f. implications of COVID-19.

The themes that guided the interviews are partly the result of a systematic literature review (Martínez-Izquierdo and Torres Sánchez 2023), and a bibliometric analysis (Martínez Izquierdo et al. 2023) of dual vocational training carried out by members of the research group "Educational Policies and Reforms" of the University of Granada and the coordinator of the research project, and in co-participation with other researchers listed as members of the project from the University of La Laguna, the University of Las Palmas de Gran Canaria, and the University of Almería. This study has also benefited from the advice and review of the tool by international researchers, practitioners, and policy experts in VET from Humboldt University, the University of Stockholm, and the University of Udine. This gives the research instrument the expert validity and reliability to collect the information necessary to achieve the research objective.

The interviews were conducted from January to June 2021 and lasted between fifty and ninety minutes. The selection of the schools was based on the information published

in the database of the Regional Department of Education and Sport. Through a protocol established by the research team, we contacted the management teams, who were informed about the research and invited to participate, and expressed their consent to do so.

The interviews were mostly conducted via Google Meet, although some were face-to-face. In all cases, the ethical standards of the research and the anonymity of the actors interviewed were guaranteed. The interviews were recorded and transcribed verbatim to facilitate the analysis of the interviewees' accounts. The process of transcription and coding of the discourses was carried out by the same members of the research team who conducted the interviews and the subsequent analysis.

A content analysis of the discourse (Bardin 1991) was carried out using a system of categories that made it possible to organise the information extracted from the interviews, with the use of the free RQDA software. The categories were derived from the research framework itself, and were defined in order to establish a common principle of classification and coding of the information among the researchers (Table 2).

**Table 2.** Categories of analysis.

| | |
|---|---|
| 1. Transitions to dual VET | Educational pathways |
| | VET (dual) versus university |
| | Academic record |
| | Offer of places in dual VET |
| | Feminisation of the higher cycle of pre-school education |
| 2. Implementation of dual VET | Theoretical and practical training |
| | Selection for access to dual training in the company |
| | Assessment of students (apprentices) |
| | Assessment of their training and learning |
| 3. Companies collaborating with dual VET | Collaboration agreements |
| | Prospector teachers companies |
| | Remuneration of apprentices |
| | Cooperation between school and company |
| 4. Access to employment through dual VET | Expectations generated |
| | Training and work experience |
| | Labour recruitment |
| | Public–private job opportunities |

Source: own elaboration.

## 4. Results

The results are organised according to the categories previously described (Table 2) above in order to find out the participants' perception of the dual vocational training modality, as it is a policy that aims to favour continuity of training and avoid early school leaving. It is also worth highlighting the link between this policy and the socio-occupational insertion of young people, insofar as the interest lies in the training that student-apprentices receive from the company, and learning if it provides them with a competitive advantage for their professional practice.

### 4.1. Transitions to Dual Vocational Education and Training

The importance of training for work is crucial in the study of educational models and their contractual relationship with the labour market. In a socio-economic scenario permeated by the economic crisis and the post-COVID-19 pandemic, the need to alternate training and work processes becomes even more appropriate, thus improving the possibilities for labour market insertion. In other words, work-based learning, focusing on more

specific and practical training, will contribute to the acquisition of competences, attitudes and skills closer to a specific job.

However, the patterns of transition to adulthood, which can be moulded according to social and economic changes, indicate transformations in the trajectories towards employment. On the one hand, the unstructured trajectories of those young people with a precarious and unstable profile, who show significant problems maintaining continuity in the same job, with constant changes and short contracts. On the other hand, there are semi-structured trajectories which, although they show more job stability and fewer changes of company, represent low-level professional categories and a lowered expectations. And finally, there are trajectories with expectations of success, determined by greater professional security and continuity, with permanent and stable contracts and with professional categories with a higher level of training and qualifications (Barrientos Sánchez et al. 2019).

### 4.1.1. Educational Pathways

Given that the training cycle on which this study focuses is at a higher level, the young students who enter the vocational training mainly come from an upper secondary education, although there are also some students who enter the vocational option once they have completed their university studies, or older students who resume their studies via this educational pathway.

> **A dual VET coordinator:** "They tend to come more from upper secondary education, a percentage from Intermediate Level. We have a percentage of older women" (School 6, public).

> **A student:** "My background in education is extensive, I am thirty-three years old, I came from a university career" (School 6, public).

### 4.1.2. VET (Dual) versus University

One of the particularities of accessing vocational training is that there are students who choose vocational training once they have finished a university degree, or older people who return to their studies. In both cases, they think of vocational training (dual) as a way of gaining employment and value the training possibilities it offers in terms of work experience.

> **A dual VET coordinator:** "We have a percentage of older women who, after getting married and having been mothers; resume their studies [...] with the aim of entering the labour market" (School 6, public).

### 4.1.3. Academic Record

Access to vocational training is contingent by the average score of the candidates' academic reports. In the higher cycle of early childhood education, for example, young students with lower academic scores may be able to access normal VET courses, while those with higher grades can access the dual programme.

> **A headteacher:** "The profile of the students is different depending on whether they are in Intermediate or Higher Level; in Intermediate Level, they tend to be students with a low previous academic record" (School 2, subsidised).

### 4.1.4. Offers of Places in Dual Vocational Education and Training Programs

Once enrolment has been completed, the availability of dual training places offered for each training cycle depends upon the characteristics of the dual training project drawn up by each school, which may include only a few places in dual training or all of them. Each project is approved by the Regional Department of Education and Sport and is renewed every year.

> **A young male student:** "I think that it is to get in [...] It is also [the case] that in my school they offered all the places in the dual [programme]" (School 2, subsidised).

**A young female student:** "For access, my Upper Secondary Diploma and average mark" (School 4, public).

Therefore, the possibility of accessing the dual system is related to the flexibility that the schools have in drawing up their own training project, which takes into account the characteristics of the vocational family and its links with the business environment, as well as the staff that teach and/or coordinate the dual system. As we explain, the schools establish the selection process for the students who enter dual vocational training based on criteria that go beyond the initial access grade for the higher-cycle VET.

**The headmistress of a public nursery school:** a limitation, the fact that all of them would like to have access to dual training, that the criteria are maintained in terms of admitting people in dual training, because they say that there was a lot of competition in terms of how to pass the processes to access dual training (School 5, public).

4.1.5. Feminisation of the Higher Cycle of Pre-School Education

In the choice of this cycle within the professional family of socio-cultural and community services, access according to differentiated gender roles is evident. In this case, the Higher Technician in Early Childhood Education is a feminised training cycle, although there are always some males enrolled because their motivation is to work professionally in the educational care of children.

**A dual VET coordinator:** "Almost all of them, 95%, if not 98 or 100% this year are girls" (School 6, public).

**A young male student:** "I was sure I wanted a dual module, in case once I started to do an internship and be with children, in case I really wanted that" (School 2, subsidised).

Gender concentration and segmentation in vocational studies are not unique to dual vocational education and training. This situation is also visible at the university level and in non-dual vocational training. Both gender concentration and gender segmentation are very significant, as they represent the beginning of occupational segregation and have important effects on labour market insertion (Consejo Económico y Social de España 2023). This study shows that highly feminised professional families, such as socio-cultural and community services (in which our study is framed), "have even higher insertion rates than some belonging to the so-called STEM fields. The low social valuation of professions related to caring for people is possibly detracting from their success in finding employment" (ibid. p. 62).

*4.2. Implementation of Dual VET*
4.2.1. Theoretical and Practical Training

The training programme to be developed is included in the dual vocational training project of each school. Based on this programme, the dual coordinator, in conjunction with the teaching staff, draws up a monthly plan of the contents and activities to be developed for each of the established modules. This plan, in turn, is coordinated with the dual company where each student has been assigned for their training, so that they can reach an agreement and approve the work schedule. Therefore, the harmonisation and coherence of actions between the educational centre and the company is fundamental for the dual training to develop satisfactorily for the apprentice.

In this training process between the school and the company, the students play a fundamental role in the acquisition of professional competences. They detail these competences through their description of objectives, contents, planning of activities, and development and monitoring of the proposed tasks, which are considered in the evaluation process of their training by the work and educational tutors.

**A young male student:** "Today I show her the activity and I tell her "well, I'm planning to do this activity", I show her the activity programme and she says "well, look, change this to make it better. The truth is that I feel one hundred percent integrated in the educational team where I'm doing my dual internships" (School 2, subsidised).

**A dual VET coordinator:** "[There is] a lot of work also on the part of the student because, apart from being there in the company, in the afternoon they have to start writing. We supervise the execution of the activity very well because you have to describe it to me, you have to send me photos, you have to send me a video. Also, the instruments, the tools you have used, the resources, the timing…" (School 6, public).

However, although there is a transversal process for the coordination of dual training, the interviewees also pointed out that this modality has limitations, in terms of the acquisition of basic content that the students do not always reach, which seem to be compensated for by the experience developed during the dual training period at the company.

**A dual VET teacher:** "I think that duality will facilitate their insertion, but it is true that there are also many limitations. In dual vocational training, it is true that there are often students who perhaps would have failed because, in terms of content, they have not reached the minimum requirements but, as I understand that vocational training is not about acquiring content, but about professionalising experiences, then of course, if they do their work well in the company they are in, of course they pass and have more opportunities than non-dual vocational training students" (School 5, public).

4.2.2. Selection for Access to Dual Training at a Company

Each school has a unique and different dual training project, which specifies the entry requirements for the selection process of students for dual training in a company. However, the projects may share aspects such as students' applications to find out their preferences about the different companies offered, visits to the centres where they can do dual training, assessment rubrics, personal interviews and academic grades for the first trimester in which they did the vocational training. In all the cases analysed, the schools carried out the selection process to assign the students to the companies, with no participation in this selection process by the work centres.

**The headmistress of a public nursery school:** "For us, the selection is made by the school [students with] the best marks are the ones that have access to dual training, we have nothing to do with that part" (School 5, public).

**A young male student:** "We had a week in which we visited the centres where we can do the dual […] you visit all the centres and then they give you a form where you have to fill in the order of preference" (School 2, subsidised).

Once the process of allocating students in the company (nursery schools) is completed, there are cases where trainees always remain at the same school, although they have the possibility of rotating through all the classes in each of the early childhood education levels. In other cases, the apprentices rotate each term through several early childhood education schools during their entire dual training period. Moreover, in other cases, during the first year the trainees conduct their dual training in public nursery schools and the second year in private nursery schools.

**A young female student:** "We were part of the morning class, we went through all the classes. The same thing, maybe, we did one week in zero years, one week in one-two and another week in two-three, and once we finished the day there, then we had two hours that we went to a dual training class" (School 5, public).

**A dual VET tutor at a school:** "We let the pupils choose in order of preference what they want, depending on their marks… In the second year, we tell the

pupils that they should look for a nursery school that especially appeals to them; we try to make them private schools so that they have more possibilities of being hired" (School 4, public).

### 4.2.3. Assessment of Their Training and Learning

Students considered the dual training experience to be very satisfactory, both in terms of the training received from the educational centre and the training received from the work centres.

**A dual VET coordinator:** "What is learnt in dual training, we cannot teach in the classroom. Contents taught in this way, in a vacuum, are forgotten. What they learn in companies is much richer than what I can teach them in the classroom" (School 6, public).

**A young female student:** "I didn't want dual training and then I loved it. The truth is that I think it's a very good way of learning because, as we know, theory in the classroom is one thing, but then, when you put it into practice, you realise many other things, you learn more and you learn better" (School 4, public).

Students are also interested in the importance of experiential learning that professionalises students through company-based training, which involves learning that is experienced and not just studied.

**A young female student:** "I mean, the difference in terms of concepts, for me it is more important to work on a concept and put it into practice, than perhaps to work on a concept and then forget about it, because if you don't work on it or put it into practice, in the end you forget about it. On a professional level, it makes you learn a lot, that can be a great advantage, you gain a lot of experience and when it comes to working, you can see it" (School 5, public).

From their dual training in nursery schools, the students emphasised the possibilities they had to observe, and to develop through practice the content they studied in the different modules. At the same time, the dual training has also enabled them to approach the different educational philosophies of educational care for children developed in public and/or private work centres.

**A young male student:** "I think that it is a school that works very well, there is a very good atmosphere, they let you do the activities that you propose, and they let you do them with the children" (School 2, subsidised).

**A young female student:** "In my case, the relationship with the teaching staff has been great. And in the case of my classmates, well, the same, I don't think any of them have had any problems with any work tutor, they have always been cooperative, both with us and with the centre" (School 4, public).

And even the public–private difference is also visible in the students' preference for dual training in a private early childhood education centre because of the future job opportunities that may arise.

**A headmistress of a public nursery school:** "Who often ask to do their internships in private institutions because it will be easier for them to enter the labour market in a private or subsidised institution than to do them here, because in the public sector it is through employment exchanges" (School 5, public).

**A young female student:** "In my case, all my training has been in private centres because I saw the possibility of a job in the future. We know that the Junta goes by employment exchange and there, no matter how much you want to, you can't get in easily" (School 4, public).

*4.3. Business Sector Collaborating with Dual VET*

4.3.1. Collaboration Agreements

For the development of dual vocational training at a company to be feasible, the functions carried out by the teachers who coordinate training in the schools are key. Among these functions, several key aspects stand out, including the initiation of contact with the companies to elaborate on the project details, to involving the work centres in the project, to selecting companies, to generating collaboration agreements, and to establishing communication between the teaching tutors and the work tutors, all of which makes it possible to present the apprentices to the assigned work centre.

> **A dual VET coordinator:** "We contact the companies to provide them with information about dual VET so that they can collaborate with us and when we have explained what the requirements of the project are, we add the companies to the project formalise all the documentation and put the teaching tutors in contact with the work tutors" (School 6, public).

To execute the process of contacting and selecting the companies to participate in the dual program, educational centres initially rely on a list of collaborating entities that offer training opportunities in work centres for students. However, the stakeholders involved in this procedure highlight specific challenges, including issues with coordination, bureaucratic inefficiencies, and the problematic approach of treating teachers like sales representatives. Moreover, there is a lack of consideration for the distinctive characteristics of early childhood education centres compared to other types of companies, which can hinder the development of effective dual training initiatives.

> **A teaching tutor:** "The initial steps involve selecting companies, and in terms of coordination within Andalusia, it seems that this aspect has not been thoroughly considered. Teachers are sometimes treated as if they were salespeople, and the unique nature of various vocational families is not always considered. For instance, an administrative internship differs significantly from an internship in a nursery school. Collaboration agreements are established only after companies have agreed to participate, but this process has encountered several challenges, particularly in the administrative aspect, which has been quite cumbersome" (School 4, public).

4.3.2. Prospector Teacher Companies

Over the past two years, the process of engaging companies to participate in dual VET programs has been facilitated through a specialized initiative called 'Prospector Teacher Companies', organized by the Regional Department of Education and Sport of Andalusia. The primary aim of this program is to identify and expand the pool of companies that can offer training opportunities within dual vocational training projects initiated by educational institutions. By fostering collaboration with the business sector, this initiative seeks to enhance the future employability prospects of dual apprentices.

> **A coordinator of a dual VET school:** "This started last year, the issue of prospecting, although I'm not prospecting because of the scholarships" (School 6, public).

While the 'Prospector Teacher Companies' initiative holds promise, it has encountered certain challenges due to its prescribed procedure. Specifically, the process of prospecting companies has been conducted outside regular school hours, leading to practical difficulties for the teaching staff involved. Moreover, the financial remuneration for their efforts is contingent upon successfully reaching an agreement, which may deter some participants. Additionally, the current requirement from the Regional Department of Education and Sport that companies remunerate apprentices in dual training programs has introduced further complexities, potentially leading to a decrease in business participation.

> **A tutor at a dual VET school:** "The initiative can indeed be intriguing, but it required conducting the work outside our regular school hours, specifically in

the afternoons. As for the compensation, I vaguely recall being paid based on the number of hours spent in meetings last year. However, during the recent prospecting phase, we chose not to participate due to reservations about the planning approach. The remuneration was solely tied to reaching an agreement, which we found problematic, especially considering the additional responsibility of securing payment for the student from the company this year" (School 4, public).

This initiative has been rejected by the teaching staff involved in the dual training projects and has meant that only a few of the schools studied have taken part in the call for prospective teachers. The educators assert that their role is primarily educational and should not involve commercial aspects. One of the teaching tutors even characterized this process as 'exploitative'.

> **A tutor at a dual VET school:** "The approach taken by the Junta with this application call was to place the responsibility on teachers to search for companies, essentially turning them into salespeople. Dealing with companies and investing time, only to be met with responses like "I want students, but I can't pay them", becomes a laborious task resulting in wasted efforts without any agreement. As a result, this year we chose not to participate as prospectors, as we believe it amounts to exploitation. Instead, we advocate for an intermediary role, someone who establishes contacts and ensures proper remuneration, thereby shifting towards a more educational-focused profile" (School 4, public).

### 4.3.3. Remuneration of Apprentices

One of the key innovations brought about by the new legislation pertains to the grant system for training contracts in the intensive vocational training regime. According to the law, "intensive vocational training" refers to a form of vocational education that involves a combination of training at vocational centres and practical experience within a company or a similar organization engaged in productive activities, with the participants receiving compensation through a training contract (Organic Law 3/2022 2022, article 67.1). Although this provision might serve as an additional motivation for young individuals in their selection of training pathways, it has also triggered apprehension and hesitation within the vocational training community of professionals.

As the participants explained in their statements, this prospecting process is being made more difficult by the current requirements, due to the fact that the Regional Department of Education and Sport has established the obligation for apprentices in training to receive grants from the companies where they carry out the dual training. This obligation has generated rejection because it incurs additional expenses and difficulties for the companies, which makes the continuity of the collaborating companies in the dual training more challenging. At this point, it is worth remembering that at the time of the interviews, the law had only recently been passed and it has not been specified how these internships are to be remunerated.

> **A dual VET coordinator:** "This issue really worries me a lot because companies are not willing to give scholarships to students. There is no specific legislation on the amount of money in the scholarships. The payment for registering students with the Social Security. But for the companies all this means a management cost that they don't have at the moment and paying that money for Social Security in advance, they have to generate a payroll, the management company is going to charge them" (School 6, public).

They consider that the processes should be made easier for the companies and that certain measures should be established to compensate the work centres that collaborate with dual training.

> **A dual VET tutor at a school:** "The Junta should play an active role with fiscal and business measures, to make it attractive in an economic way to companies [. . .] or maybe instead of them having to pay, to reduce taxes" (School 4, public).

In addition, the requirement to remunerate students in dual training only applies to private and/or subsidised centres and not to public nursery schools. Therefore, the difficulty of offering all the places in dual training will be a major challenge for the educational centres that already offer dual VET in early childhood education. This situation is aggravated by the fact that students who have completed their dual training in public schools do not have the possibility of being hired when they finish their training, as public schools depend on the territorial government and do not have the competence to hire.

> **A dual VET tutor at a school:** "It has been impossible for any company to pay the internships [to those who have already been trained]. It will be even more difficult for them to pay people who have only been training for three months in the dual programme. Moreover, another very curious thing is that in the world of nursery schools, we have many public schools that belong to the Junta, so the Junta does not need the Junta to pay but a private school that may only have three workers; we are forcing them to pay" (School 4, public).

### 4.3.4. Cooperation between Educational Centres and Companies

In spite of the limitations mentioned above, the respondents considered that the collaborative role of the companies (public, private, subsidised by the regional educational administration, and SMEs) is fundamental in order to provide training and learning spaces for future workers. This is especially true if the goal of the dual system is to favour cooperation between the educational and labour sectors with the aim of improving the integration of young people into the labour market. In fact, in countries where dual vocational training is consolidated, youth unemployment is relatively low, insofar as training is linked to the labour market and students learn to perform professionally in the practice and culture of business (Alemán-Falcón and Calcines-Piñero 2022).

> **A dual VET coordinator:** "[The companies] really do it because they believe that they have to be given an opportunity to learn, because learning to work with people takes place with people. They do it as a social value to encourage learning and to encourage them to enter the labour market" (School 6, public).

In fact, they state that there is a positive collaboration with the companies, insofar as they cooperate and interact in everything related to training. This includes the planning of regular meetings between the teaching tutor and the work tutor to attend to the monitoring of the training that the apprentices are carrying out at the company, as well as the the grading of the young students in dual training.

> **A dual VET coordinator:** "The coordination with the companies is fantastic, I think they say yes to everything". (School 4, public).

> **Headmistress of a public nursery school:** "She has her regular visits both with the tutor and with me to see how everything is going, to see if there are any problems [. . .] that is, there is a scheduled follow-up [. . .] In fact, there is a register where the visit is signed and stamped" (School 5, public).

### 4.4. Access to Employment through Dual VET

### 4.4.1. Expectations Generated

Initially, studying in a dual mode seems to provide apprentices with a more competitive level of training due to the practical experience received, the specialisation in an area of work, and the acquisition of professional skills. These are conditions that dual vocational training seems to fulfil as a training policy. Consequently, this setup has evolved into a training and work-based learning environment that enhances the transition from school to the workforce. It cultivates a more intimate and direct connection between the education system, businesses, and the production model, ultimately nurturing stronger ties with diverse professional profiles (Barrientos Sánchez et al. 2019).

> **A young male student:** "I think that a person who does the dual module comes out much better prepared for the world of work than a person who does the traditional module, although that person also has four months of work experience afterwards. But those who do the dual module have six months in the first year, another six in the second year, plus the four months" (School 2, subsidised).

### 4.4.2. Training and Work Experience

Certain informants asserted that the fundamental objective of vocational training is to gain practical experience, facilitating entry into the labour market. Consequently, they emphasized the importance of learning through actual employment in a role that aligns with their desired job, immersing themselves in a dedicated work environment—a circumstance effectively facilitated by dual training.

> **A dual VET coordinator:** "Is that at the end of the day, vocational training is labour insertion, so you have to learn to work by working" (School 6, public).

Indeed, students who have successfully completed the training have pointed out that the dual mode of vocational education is also evident in the documentation accompanying the Higher Technical Degree in Early Childhood Education. This documentation includes a formal record from the work centre, explicitly stating the hours dedicated to the practical dual training component of the program.

> **A young female student:** "Yeah, when we complete the cycle, on the document, it shows the overall grades and all that stuff, but it also mentions that it was done in the dual mode. Besides, they give us this paper from the work centre showing the hours we spent there, no matter if it's a public, private, or subsidized centre" (School 5, public).

### 4.4.3. Labour Recruitment

Indeed, there is unanimous agreement that dual training confers a competitive advantage to individuals seeking employment in the labour market; a phenomenon observed on several occasions after the completion of training at the early childhood education centre. However, it should be noted that not all the students who complete dual education secure employment upon completing their training.

> **The director of a private company:** "Because we have already seen how they work we tend to hire individuals who have undergone dual training. In fact, last year, we hired a young woman right after she completed her dual internship to cover a temporary position" (School 2, subsidised).

> **A dual VET tutor at a school:** "We encourage students to consider private companies for their dual training as it enhances their chances of getting hired. Some companies have already expressed interest in hiring our students. However, we'll have to wait a few months to see how things unfold" (School 4, public).

### 4.4.4. Public–Private Job Opportunities

There is also agreement on the statement that the probability of easier access to employment is made possible by dual training in private workplaces, since public early childhood education centres cannot hire their staff directly.

> **Headmistress of a public nursery school:** "I can tell you about the ones who left last year in their second year who are already working. One of them is in a subsidised school and another one is in our school, because as there is an external company, which is in charge of the holiday periods. I always tell them that until they get a job in the public sector, they should go to the private-concerted [semi-private] sector, to get points" (School 5, public).

For this reason, some schools make it possible for all students to complete at least one period of in-company training in privately owned workplaces, as this offers more opportunities for future employment.

> **A dual VET tutor at a school:** "We tell the students that they should look for a nursery school that particularly appeals to them; we try to make sure that they are privately owned so that they have more opportunities for recruitment in terms of training in the workplace" (School 4, public).

## 5. Discussions and Conclusions

The present study specifies the existence of a number of particular characteristics in the approach to dual vocational training in Spain. On the one hand, there has been an attempt to import a training policy characteristic of the German model without taking into account a needs analysis and the possibilities that would allow for its adaptation to the Spanish context for its development (Martínez-Izquierdo and Torres Sánchez 2023). This has been actively promoted by the EU and the OECD as a valuable training option for young people's transition to work and as a way of acquiring skills for labour market integration (Šćepanović and Martín Artiles 2020).

On the other hand, although the dual VET has been regulated by law, the path developed over almost a decade has been carried out without a defined framework and with the presence of companies and foundations as influential networks for its implementation (Barroso-Hurtado et al. 2021). This situation has led to varied interpretations and implementations by the autonomous communities (Marhuenda-Fluixá et al. 2019), which must comply with the recently approved Spanish law on the organisation and integration of VET in order to deliver dual education (Organic Law 3/2022 2022).

In this context, in the autonomous community of Andalusia, in accordance with the annual call of the Regional Department of Education and Sport, each school presents its own dual training project. Although the call specifies the requirements for dual training projects, in practice, the development of the projects may be different for the same cycle of a professional family. This is the case, for example, when dealing with issues such as access criteria to dual training, the theoretical–practical training programme, the selection procedures and criteria established for the assignment of training positions in a company, or the evaluation of the apprentices from the work centres, among others. These elements are key for the implementation processes of a dual VET program, and have been shown in the literature as a series of difficulties to be faced in the proper development of a dual VET program (Martínez-Izquierdo and Torres Sánchez 2023).

The research from an international perspective by Valiente and Scandurra (2017) concludes that problems are generated when transferring global policies to local contexts, insofar as it is necessary to know how these training models interact with the social, educational, and economic contexts in which they are developed, and that these policies do not reach a systemic level of development. The main challenge is to make dual apprenticeships attractive to young people and the business sector, the latter being key to the implementation of dual training. In this sense, it is important to investigate the relationship between the theoretical–practical training of the curriculum at schools and the applied curriculum, in order to know the learning achievements possible at the companies; this was established by Cedefop's (2022) research as one of the concrete actions for further research.

Despite certain bureaucratic and organisational limitations, the accounts of the participants in the dual training program of the Higher Technical Diploma in Early Childhood Education showed a pragmatic assessment of the relevance of dual vocational training. This is particularly true as the training developed at the work centres emphasises the importance of professional practice and the connection between the theoretical–practical contents, despite the fact that early childhood education centres have unique characteristics as companies. Moreover, this training process contributes to the professional certification of apprentices and makes it possible to update their work skills, even though companies

are sometimes unaware of the relevance of their functions as apprentices within a dual training process.

The positive evaluation of dual vocational training gains particular significance when considering its potential role as a policy instrument with which to combat the persistently high rates of youth unemployment in Spain (Barroso-Hurtado et al. 2021). On this issue, the actors involved consider that completing dual vocational training entails a greater probability of labour market insertion. This arises because of the professional experience that dual vocational training offers through training in work centres, especially if the placements are at private early childhood education centres. In this sense, Homs (2023) emphasizes the importance of dual VET for training and not as a way to find a job, "although it is likely that at the end of your training the company will offer you a job" (p. 198).

However, these expectations could face challenges due to the current requirements set forth by the Regional Department of Education and Sport concerning the "teacher prospector companies". This commitment demands that educational institutions offering dual education must partake in this call, which aims to expand and increase the number of participating business involved in dual education. In this sense, it is worrying to know the way in which this process of involving companies has been approached, opting to send teachers as "sales agents" to sell them an idea that should have been put forward by the business sphere and not by the Ministry of Education (Marhuenda-Fluixá et al. 2019, p. 218).

This process of involving businesses becomes even more challenging, as under the current law, private companies are obligated to provide compensation to apprentices, a fundamental feature of the German dual training system (Alemán-Falcón and Calcines-Piñero 2022). In Spain, only a few regions require a contract to participate in dual vocational training, and most accept a scholarship instead of a contract or do not even require any payment for the student. Therefore, the student lacks the status of an apprentice, even if enrolled in dual vocational training (Marhuenda-Fluixá et al. 2019).

In the case of the higher cycle of early childhood education, dual training takes place in both public and private schools. However, for private educational institutions, the potential implementation of student remuneration diminishes and, in some cases, completely inhibits their participation in dual training. Paradoxically, this conflicts with the fact that private centres offer the highest probability for dual training apprentices to secure employment opportunities upon the completion of their training.

Similarly, Brunetti and Corsini (2019) argue for better employability in both the short and medium term for young people who have studied in a dual vocational training program, i.e., those countries where this type of training is implemented and works in an organised way with the labour market, have better success results than in countries with a school-based vocational training system, which have mixed results, and where only in some cases is the impact of these studies significantly positive. In the case of Spain, the research by Martín Artiles et al. (2020) indicates that "the results show that dual VET has been implemented through a school-based model, as opposed to the firm-based 'German' model" although "whichever approach was taken, we found that young people who have completed dual VET enjoy a rapid school-to-work transition" (p. 73).

Consequently, based on what has been investigated, it is a priority to continue research in several directions: the implications of the new regulation of this educational policy in the reduction in early school leaving; carrying out a follow-up study of young students who have graduated from dual training for the Higher Technician in Early Childhood Education, in order to find out about their career path and assess to what extent dual training represents a competitive advantage for labour integration; studying the assessments of the companies receiving the young people undergoing training in this modality, and the implementation of the same; and also to know the relevance or not of implementing dual vocational training in the higher vocational training cycle in early childhood education, due to the particularities of this training.

These research prospects are set out in our current research project, "Connecting learning and meaningful work in Andalusia: comparative research on dual vocational

training in Early Childhood Education" (PI21_00162), funded by the Andalusian Plan for Research, Development, and Innovation of the Andalusian Regional Government (Junta de Andalucía). We are motivated to delve into the learning experiences that the dual mode of vocational training brings to early childhood education in Andalusia. In particular, one of the objectives is focused on studying the transitions from the training system to the labour market made by young people in dual vocational training in early childhood education, through a quantitative study that will allow us to provide an overview of the Andalusian context.

**Author Contributions:** Conceptualization M.J.R., R.L.G. and J.G.F.; methodology, M.J.R. and R.L.G.; formal analysis, M.J.R., R.L.G. and J.G.F.; investigation, M.J.R. and R.L.G.; resources, M.J.R., R.L.G. and J.G.F.; data curation, M.J.R.; writing—original draft preparation, M.J.R., R.L.G. and J.G.F.; writing—review and editing, M.J.R., R.L.G. and J.G.F.; visualization, M.J.R.; supervision, R.L.G.; project administration, M.J.R.; funding acquisition, M.J.R., R.L.G. and J.G.F. All authors have read and agreed to the published version of the manuscript.

**Funding:** This research was funded by the Ministry of Science and Innovation of the Government of Spain; grant number RTI2018-101660-B-100.

**Institutional Review Board Statement:** The ethical principles of research are authorized through the Research Project financed by the Ministry of Science and Innovation of the Government of Spain; grant number RTI2018-101660-B-100.

**Informed Consent Statement:** Informed consent was obtained from all the subjects involved in this study.

**Data Availability Statement:** Data has been curated and processed following MDPI Research Data Policies available at https://www.mdpi.com/ethics (accessed on 5 June 2023).

**Conflicts of Interest:** The authors declare no conflict of interest.

**Appendix A**

**Table A1.** List of interviews.

| Autonomous Community of Andalusia | | | | |
|---|---|---|---|---|
| **Province** | **Position** | **Sex** | **School** | **Type of Education Institutions** |
| MÁLAGA | Headteacher at an educational centre | Male | 1 | Subsidised |
| | Work tutor | Male | | |
| | Young student | Female | | |
| | Young student | Male | | |
| | Director of a private company | Female | | |
| SEVILLA | Headteacher at an educational centre | Male | 2 | Subsidised |
| | Coordinator of a dual VET school | Female | | |
| | Young student | Female | | |
| | Young student | Male | | |
| | Director of a private company | Female | | |
| | Director of a public company | Male | | |
| | Work tutor at a public company | Female | | |

**Table A1.** *Cont.*

| | Autonomous Community of Andalusia | | | |
|---|---|---|---|---|
| **Province** | **Position** | **Sex** | **School** | **Type of Education Institutions** |
| GRANADA | Vice-Headteacher | Male | 3 | Public |
| | Coordinator of a VET school | Female | | |
| | Young student | Male | | |
| | Director of a private company | Female | | |
| CÁDIZ | Coordinator of a dual VET school | Female | 4 | Public |
| | Tutor at a dual VET school | Female | | |
| | Young student | Male | | |
| | Young student | Female | | |
| | Young student | Female | | |
| ALMERÍA | Coordinator of a dual VET school | Female | 5 | Public |
| | Tutor at a dual VET school | Female | | |
| | Young student | Female | | |
| | Young student | Female | | |
| | Director of a private company | Female | | |
| | Headmistress of a nursery school and Director of a public company | Female | | |
| CÓRDOBA | Coordinator of a dual VET school | Female | 6 | Public |
| HUELVA | Coordinator of work tutors | Female | 7 | Private |

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
