# Peer review of "Dual Vocational Education and Training Policy in Andalusia: The Nexus between the Education System and the Business Sector in the Higher-Level Training Cycle of Early Childhood Education"

_socsci, doi:10.3390/socsci12090519_

Round 1
Reviewer 1 Report
The subject matter appears to be of particular relevance and usefulness to the Spanish education system. This entails an initial reflection on the paper presented: of what use is it to a much wider audience, which is what an international English-language scholarly journal is targeting? If we look at the bibliography cited, it too is composed of papers almost seclusively in Spanish and referencing Spanish research and, very often, non-scientific papers. I think the paper is better suited to journals that are more focused on both topic and country, such as Archivos Analíticos de Política Educativa, and Revista de Investigación Educativa.
The presentation of many non-scientific papers suggests reflection on the type of product. The general impression is that we are not dealing with a scientific paper but with a "report," which as such does not have to follow specific rules. Of these I mention one, which is to choose, motivate, describe and use specific procedures for data collection and analysis. Since in this case we are referring to qualitative research conducted with interviews, it turns out that the procedure for coding and analyzing data is completely inappropriate for a scientific journal. No specific software such as NVivo was used, but this is optional, but no coding mode and a procedure involving multiple coders is indicated, leaving ample room for the risk of interpretation being filtered and driven by the author's knowledge and pre-judgments. Saying "A content analysis (Bardin, 1991) was conducted, and relevant themes described by the informants were categorized for further examination." is wholly insufficient for a scholarly journal.
Author Response
Dear reviewer:
Thank you for the comments made in the review of the article. In response to your comments, we would like to point out the following:
1.- The implementation of dual vocational training policy in Spain, as in these EU and OECD countries, is developing as a consequence of the transfer of global policies to national and regional contexts. In this sense, the review of the article has been based on this issue, which shows how the application of global policies in specific contexts entails a series of risks and difficulties, due to the different historical, social, cultural, educational and economic trajectories in each country. We have provided English-language literature that highlights these issues. And all bibliographical references are to edited books, book chapters and journal articles. There are no "non-scientific" articles.
We present a research from the context of Andalusia focused on the development of dual vocational training as a "travelling policy" that at the request of the European Union, different regulations and the Ministry of Education and Vocational Training in Spain, has been taken as a reference to address early school leaving and the greater likelihood of labour integration of young people.
2.- The methodology is qualitative because this is endorsed in the research project financed by the Ministry of Science, Innovation and Universities. Qualitative research is another way of studying reality in the field of Social Sciences. The fact that it is qualitative does not imply that the research is not scientific or a report. This approach was chosen in order to meet the objectives of the project. We have expanded the methodology section in depth. In the current research project funded by the regional government, we are going to study the transitions of young people from dual vocational training to the labour market with a quantitative perspective.
Best regards.
Reviewer 2 Report
Paper gives a good description of the evolution of the dual training system in Spain. The findings, however, are anecdotal because they are based on a few interviews of stakeholders. A potential revision of this paper should include a review of the German dual system and its effect on early school leaving, employment and earnings. Then a tracer study should be conducted of recent graduates. The analysis should be quantitative and not just descriptive.
Author Response
Dear reviewer:
Thank you for the comments made in the review of the article. In response to your comments, we would like to indicate the following:
1.- We present a research from the context of Andalusia focused on the development of dual vocational training as a "travelling policy" that at the behest of the European Union, different regulations and the Ministry of Education and Vocational Training in Spain, has been taken as a reference to tackle early school leaving and the greater likelihood of labour integration of young people. Therefore, our aim is not to offer a description of the dual vocational training system in Germany, although we have gone in depth in the review of the article to highlight the most significant aspects of the dual mode in Germany, providing specific bibliographical references.
The implementation of the dual vocational training policy in Spain, as in these European Union and OECD countries, is developing as a consequence of the transfer of global policies to national and regional contexts. In this sense, the review of the article has been based on this issue, which shows how the application of global policies in specific contexts entails a series of risks and difficulties, due to the different historical, social, cultural, educational and economic trajectories in each country. We have provided English-language literature that highlights these issues.
In the conclusions, we have highlighted the influence of the transfer of dual vocational training policies from other countries (Germany) on the educational and employment context in Spain.
2.- In the article we have added an appendix with the table of the interviewees. In order to have a broader view of how dual vocational training is being implemented, it is key to look at the different actors involved in the development of the policy. And among these actors there are 10 young people interviewed. They were indicated as "student" and now we have indicated "young student". Young people are the beneficiaries of the policy, but they are not the only ones involved in its implementation. Regarding the number of interviews, it indicates that there are "few interviews". The interviews are what they are because at the time the interviews were carried out, there were very few educational centres offering dual vocational training in early childhood education. In fact, there were some provinces in Andalusia that did not even offer dual vocational training in early childhood education. At present, there are not a relevant number of schools either. We have expanded and described this information in the article.
3.- As you have told us, in the current research project funded by the regional government, we are going to study the transitions of young people from dual vocational training to the labour market with a quantitative perspective.
Best regards.
Round 2
Reviewer 1 Report
The article has been improved, but I still think it would be better placed in journals specifically dedicated to this type of topic.
Author Response
Dear reviewer:
Thank you very much for your comment. We will take it into consideration for a future publication.
Best regards.
Reviewer 2 Report
The paper will benefit from a tracer study to assess the transition of dual vocational training graduates to the labor market, as compared to a control group of non-vocational graduates.
Author Response
Dear reviewer:
Thank you for your comment. However, we would like to point out that our research objective is different from the observation you indicate.
Best regards.